# Financialisation of Housing in London: Empirical Evidence on Housing Prices

**José Francisco Vergara-Perucich**

Centro Producción del Espacio, Universidad de Las Américas, Santiago 7500975, Chile; jvergara@udla.cl

**Abstract:** This paper aims to empirically review the process of housing financialisation in London, exploring a time series causal relationship between house prices and financial instruments, using the Granger method and a VAR test. In order to carry out this analysis, we use a vector autoregressive model with a monthly data series that seeks to contribute to exploring this relationship. The results are relevant to the important role that the theory of housing financialisation plays in explaining the crisis of access to secure tenure that can be seen in developed nations. The results also provide an empirical background to pursue this theory more specifically in the context of the vectors that are effectively causal to the financialisation processes that impact everyday life through housing prices. The study is original, given that this type of modelling has not previously been carried out for a major world city such as London, and adds to the findings of similar explorations that have applied other methodologies.

**Keywords:** financialisation; housing prices; London; FTSE 100; central bank

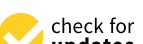



## 1. Introduction

The process of housing financialisation has direct effects on the fair allocation of housing in the world's cities. This process of financialisation is partly framed by the granting of debt as the main mechanism for accessing home ownership, a mortgage process in which financial institutions are directly involved as guarantors of the volume of money needed to acquire housing [1,2]. The financialisation model is a reproductive factor of spatial inequality, as it distances secure access to housing even further from the decision-making power of households and of the State itself, which is constrained to comply with the demands of financial institutions to grant credits that allow the population to access housing [3]. It appears that lessons have not been learnt about the extremely close relationship between financial systems and access to basic human rights such as housing. It has already been shown that the excessive growth of mortgages was one of the main causes of the global financial crisis of 2009 [4,5]. As Aalbers points out, the financialisation of housing is a broad conceptual framework, with a certain ambiguity in its definitions that allows for broader discussions about its implications for everyday life and for variation in the theoretical frameworks that can emerge from its analyses [2]. On the other hand, some authors argue that there is a problem with this excessive openness, since the broadness and vagueness of the term could lead to misinterpretations of its real scope [6,7]. This article seeks to guide this discussion by analyzing empirical variables that allow us to situate the discussion on financialisation based on arguments validated by applied statistics. On the one hand, reference is made to the effects of financialisation processes on everyday life; however, evidence to support these arguments has not necessarily been ordered based on validated methodologies from the economic sciences in order to determine whether there are potential causal relationships between factors. In concrete terms, how is the impact of financialisation represented in people's daily lives? In this case, it is suggested that in neoliberal societies with a strong emphasis on the free market economy, the representation of

this impact can be seen in the price of housing, where the influence of financial factors on this price would allow us to measure the weight that financialisation has on everyday life.

It is important to place this assumption of price as a financialisation effect in the broader context of the literature. The first step is to define what can be understood by financialisation, for which there are various definitions. Natascha van der Zwan points out that financialisation studies have been characterised by inquiries into how finance at the global level tends to alter the industrial logic of economies at the local level in democratic societies around the world [8]. Servaas Storm argues that financialisation implies the presence of financial actions in social sectors where it was previously absent, thereby expanding the presence of rent-seeking practices that feed the wealth of the global elite, employing a strategy of depoliticizing the economy, thereby aligning the social-economic process with what Hayek claimed was the only reliable way to create welfare and social order while generating economic progress [9]. A broader definition of financialisation is given by Stockhammer, who says that it is a process in which non-financial entities begin to assimilate their activities as if they were financial or begin to develop their activities based on investment or loans from financial institutions, thus generating a certain level of dependence on these companies [10].

The process of financialisation involves a new regime of power relations in which the State operates, led by financial institutions, in a post-Fordist phase based on the deregulation of these institutions in a way that mainly follows the expansive North American policy, together with the belief that, in the neoliberal context, the economy would greatly benefit from the expansion of the global financial sector [11,12]. Financialisation, in terms of tangible effects, can be seen in the increase of income in favour of creditors and the reduction of workers' purchasing power in the face of a dual process of reduced labour productivity and increased operating costs of economies financed mainly by global financial institutions [13]. One of the common factors in the study of housing financialisation processes centres on the State's reluctance to participate in housing production and property management processes, and continued reliance on these entities despite their having caused the subprime crisis [14]. For Emma Mawdsley, there is a contradiction in reviewing the geographies of development between the evident effort by governments and non-governmental entities to financialise communities in the name of development and the fact that these same processes activate and deepen the effects of speculation on everything that ends up being converted into financial assets, including the very humanitarian causes they seek to address [15]. For Sebastian Kohl, the aggregate evidence from 13 nations that have financialised their housing policies indicates that the liberalisation of financial markets ends up being a poor substitute for more traditional models of housing policy based on strategies associated with public housing, as an example [16].

To examine the potential causal effect between financial variables and housing prices, this manuscript applies Granger causality and vector autoregressive (VAR) models. While the Granger causality model has been discussed in the literature as a process that indicates statistically influential relationships rather than causal relationships, it is widely used to verify whether one variable can influence another over time [17–19]. Thus, in this paper, when there is a mention of causal relations, it is based on Granger test results. On the other hand, VAR models allow for testing the time-based influence relationship between multiple variables [20–22]. Various studies have used these modelling strategies to examine causal relationships between variables and housing prices, starting in 1993 with Sarker's study on the relationship between Canadian wood exports and housing prices in the United States [23]. Zhang et al. use VAR-based modelling to test the effects of macroeconomic variables on housing prices, identifying a strong relationship between housing prices and interest rates [24]. Yang and Pan identify an inverse effect between housing prices and economic development for 31 provinces in China [25]. In a study conducted in Malaysia, Ibrahim and Law apply a VAR model to identify the long-term relationship between bank credits and terraced house prices [26]. D'Albis et al. identify that, in the case of France, immigration does not have a significant effect on housing prices, but high housing

prices appear to discourage migration [27]. In the field of urban studies, most of these analysis techniques have been applied in Asian nations, while the majority of economic studies take place in the United States. Kuethe and Pede demonstrate, using a VAR model and a Granger causality test, that housing prices are sensitive to macroeconomic shocks [28]. Ambrose et al., using the case of Amsterdam with a time series of 355 years, examine the relationship between fundamentals and housing prices, demonstrating that changes in the value of fundamentals affect sales prices, not rental prices, primarily [29]. Sá et al.'s study on OECD countries indicates that mortgage markets have an upward effect on housing prices [30]. There is abundant literature that uses VAR and Granger models to evaluate housing prices, but no prior studies have situated their analyses on the bases of the critical theory of financialisation of housing. The financialisation of housing has been studied through approaches that seek to identify the effects of financial capitalism on the process of production and generation of value in access to housing, from a political economy analytical framework. Thus, the financialisation of housing is constructed with the contributions of David Harvey, Raquel Rolnik, Manuel Aalbers, Neil Smith and Nancy Fraser, among many others, mostly under neo-Marxist interpretations of socio-spatial relations. By taking an econometric approach to analysing potential contradictions, this article allows the engagement with these views of other authors who, in many cases, avoid the neo-Marxist critique of the contradictions of capitalism and focus solely on the effects on housing price elasticity. In this case, the findings and methodology, as well as London's own case, allow us to bridge a communication gap between the two disciplinary approaches to applied political economy.

Through this econometric research, the article seeks to contribute to the empirical connections that emerge from the evidence that allows us to delimit in part how the global financial spheres do have an impact on aspects of local economies [31], as is the case of house prices in one of the world's major cities, London. For this city, there is not much evidence linking financialisation processes with house price formation from a time series econometric methodological approach. Some recent analyses indicate that factors such as gentrification, demand displacement, immigration, foreign in-migration and crime are relevant variables for understanding house price formation when applying a geographical convergence analysis [32]. London has been facing a housing affordability crisis for some time now, which has been reflected in a significant increase in homelessness, a reduction in new home ownership and a steady rise in house prices despite the difficulty most people have in affording them [33]. In London, housing has become a savings vehicle for large capital, while also complementing the reduction of state involvement in housing production and management [34]. Consequently, these actions result in the perpetuation of a housing access crisis that increases in severity over the years.

## 2. Materials and Methods

The dataset used for this analysis consists of monthly changes for series starting in January 1999 and ending in July 2022. A variety of data sources were used for the sample. Firstly, the official daily series from Her Majesty's Transaction Registry was used to obtain information on the value of residential property transactions in London. The City of London has been excluded from this analysis, as it is a special area within the UK in which other types of economic relationships operate that are not addressed in this paper. After excluding the City of London, two factors are included in the analysis: average house prices and the number of monthly sales. The variable to be explained in this research is, precisely, the price of housing, so its inclusion is elementary. On the other hand, the volume of monthly sales operates as a control variable in the sample, since it is expected that an increase in housing demand will have some degree of impact on the price, as mandated by general economic theory [35]. This paper aims to review, using Granger's approach, whether there are causal relationships between financial variables and house prices in the following direction of causality:

Financial Variables -> Housing Prices.

Some widely validated financial variables have been selected to conduct this exploration, considering two agents: central banks and the international stock market. On the one hand, it is understood that central banks enact monetary policy to influence the internal economic balances of each nation. In this case, the monetary policy rate of the central Bank of England has been used as an explanatory variable. Considering that the United Kingdom in general and London in particular have a great openness to international trade, the monetary policy rate of the eurozone, China and the United States of America was incorporated into the analysis. Thus, in addition to looking exclusively at the influence of territorially based economic decisions in the UK, this analysis also assesses the potential impact of the monetary policies of some of the UK's main economic partners. In addition, a purely financial factor has been incorporated, which is the daily valuation in monthly averages of the FTSE 100 Index, which is composed of the financial valuation of the main UK companies on the international stock exchange. The data series is taken with its original values, which are then transformed to work with the variations between periods, to facilitate the comparison of the dynamics between the variables used. The descriptive statistics of the variables used can be seen in Table 1. The data series were first-difference transformed to unify the sample work and achieve a unit root test suitable for time series modelling. In this sense, we work with the first difference of average house prices in London, excluding the City of London for the reasons explained above. The descriptive summary of the transformation of the data series is shown in Figure 1.

**Table 1.** Descriptive statistics of the sample. Source: Own elaboration based on Bank of England, OCDE, Yahoo Finance and Her Majesty's Transaction Registry.

| Variables | Mean | Median | Std. Dev. | Kurtosis | Range | Min | Max | Obs. |
|---|---|---|---|---|---|---|---|---|
| Housing Price | 0.0059 | 0.0056 | 0.0095 | 1.1758 | 0.0649 | −0.0305 | 0.0344 | 282 |
| Monetary Policy Rate UK | 0.0049 | 0 | 0.1220 | 19.9913 | 1.5653 | −0.6563 | 0.9091 | 282 |
| Monetary Policy Rate US | 0.0028 | 0 | 0.1289 | 28.6296 | 1.6670 | −0.7124 | 0.9545 | 282 |
| Monetary Policy Rate Eurozone | −0.0037 | 0 | 0.1382 | 24.6881 | 1.8636 | −1.0000 | 0.8636 | 282 |
| Monetary Policy Rate China | −0.0012 | 0 | 0.0163 | 35.6456 | 0.2091 | −0.1607 | 0.0484 | 282 |
| FTSE 100 | 0.0013 | 0.0046 | 0.0360 | 6.5743 | 0.3020 | −0.2149 | 0.0871 | 282 |
| Housing Sales | 0.0241 | 0.0097 | 0.2445 | 39.2634 | 3.3612 | −0.8879 | 2.4733 | 282 |

To search for causal relationships between time series using Granger's approach, the data must be appropriate for these calculations. In this case, an Augmented Dickey–Fuller test was applied to evaluate the unit root of the series to be used. Table 2 shows the results of the test, where the TAU statistic should be negative and far from zero, as has been shown in this case, while the $p$-value should be as close to zero as possible in order to reject the null hypothesis that there is a unit root in the model. This is not the case, indicating that the sample is appropriate for use in the study.

Next, in order to assess the type of evaluation to be carried out, it is necessary to identify the number of lags to be incorporated into the evaluation. For this, a selection model assisted by Gretl software is applied, which helps to establish how many lags are necessary to perform a VAR model analysis for a given time series. Table 3 shows the results of this study model, applying the Akaike criterion, which indicates that at least 3 lags should be incorporated into the model, given that this is the most relevant result after analysing the series in 8 individual lags.

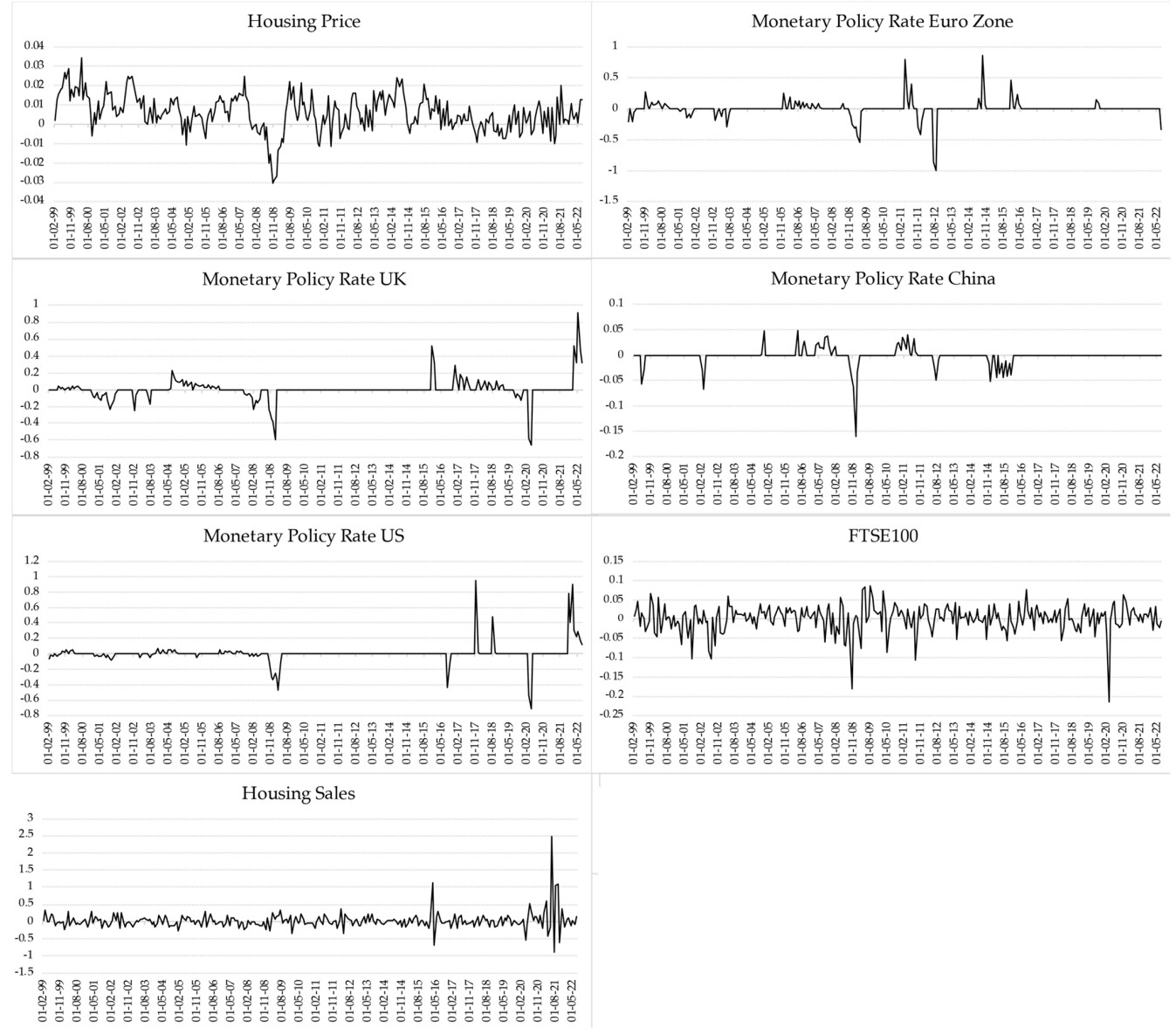

**Figure 1.** Variables in used to run the analysis. Source: Own elaboration based on Bank of England, OCDE, Yahoo Finance and Her Majesty's Transaction Registry.

**Table 2.** Augmented Dickey–Fuller test results. Source: Own elaboration.

| Variables | *p*-Value | Estimated Value of (a-1) | Test Statistic (tau_nc (1)) | 1st-Order Autocorrelation coeff. for e | N |
|---|---|---|---|---|---|
| London House Price (average) | $8.15 \times 10^{-5}$ | −0.157 | −394.364 | 0.025 | 280 |
| FTSE 100 | $3.33 \times 10^{-26}$ | −0.987 | −125.252 | 0.005 | 280 |
| Monetary Policy Rate England | $6.03 \times 10^{-13}$ | −0.383 | −7.972 | −0.030 | 280 |
| Monetary Policy Rate United States | $4.68 \times 10^{-11}$ | −0.452 | −7.310 | −0.002 | 280 |
| Monetary Policy Rate Eurozone | $7.81 \times 10^{-10}$ | −0.561 | −6.861 | −0.001 | 280 |
| Monetary Policy Rate China | $7.90 \times 10^{-12}$ | −0.459 | −7.585 | −0.005 | 280 |
| House Sales | $5.66 \times 10^{-11}$ | −0.853 | −7.280 | −0.020 | 280 |
| City of London House Price | $7.30 \times 10^{-36}$ | −1.248 | −15.236 | 0.010 | 280 |

**Table 3.** Lag order selected by the Akaike criterion. Source: Own elaboration.

| Lags | loglik | p(LR) | AIC |
|------|--------|-------|-----|
| 1 | 2967.36468 | −21.250837 | −20.512388 |
| 2 | 3035.66885 | 0 | −21.391743 |
| 3 | 3112.02178 | 0 | −21.591400 * |
| 4 | 3150.258 | 0.00726 | −21.512832 |
| 5 | 3190.33673 | 0.00328 | −21.447713 |
| 6 | 3219.37166 | 0.1758 | −21.301983 |
| 7 | 3250.94774 | 0.08422 | −21.174801 |
| 8 | 3279.981 | 0.17587 | −21.029058 |

* = most statistically significant result.

The modelling of the causal relationships between the variables will be carried out in two instances. The first will be exploration by applying the Granger causality test. This test indicates whether there are potential causal relationships in the Granger sense between two variables and the direction in which this causality occurs [17,36]. In other words, the test allows us to see whether house prices are causing the increase in sales or vice versa, or whether both causal relationships are occurring at the same time. The Granger causality test is constructed from a set of bivariate regressions at various relational moments between the variables, according to the number of lags that the study contains [37], from the following general formula combining x with y:

$$y_t = \alpha 0 + \alpha 1 y_{t-1} \ldots + \alpha 1 y_{t-1} + B_1 x_{t-1} \ldots B_i x_{-i} + \epsilon_t \tag{1}$$

$$x_t = \alpha 0 + \alpha 1 x_{t-1} \ldots + \alpha 1 x_{t-1} + B_1 y_{t-1} \ldots B_i y_{-i} + u_t \tag{2}$$

The null hypothesis is that variable X does not cause Granger variations in a variable Y in the Granger sense. This econometric technique will allow us to identify, mainly, the direction of the relationships between the variables associated with the price of the house under study. Having established these results, we can then review the vector autoregressive analysis that will allow us to look simultaneously and comparatively at the influence that the explanatory variables may have on the variable to be explained, in this case, the price of housing in London. VAR modelling seeks to identify the persistence over time of one vector over the other for a multivariate time series. The equation for estimating a VAR model is as follows:

$$y_t = A_1 y_{t-1} + A_2 y_{t-2} + \cdots + A p y_{t-p} + B x_t + \epsilon_t$$

where p is the order of the VAR, identifying in this factor the number of lags the model has. On the other hand, Xt can be used as an exogenous variable and $\epsilon_t$ is a white noise vector. The polynomial matrices are ordered in a similar way to a re-regression, but incorporating the lags, as can be seen in the following equation:

$$\begin{bmatrix} Yt \\ Yt-1 \\ \ldots \\ Yt-p-1 \end{bmatrix} = A \begin{bmatrix} Yt-1 \\ Yt-2 \\ \ldots \\ Yt-p \end{bmatrix} + \begin{bmatrix} B \\ 0 \\ \ldots \\ 0 \end{bmatrix} Xt + \begin{bmatrix} \epsilon t \\ 0 \\ \ldots \\ 0 \end{bmatrix}$$

A synthetic equation is written as follows:

$$A(L)y_t = Bx_t + \epsilon_t$$

The results of the application of this model are presented below.

## 3. Results

First, we present the results of the Granger causality tests (Table 4), which indicate that there are statistically significant (*p*-value < 0.05) Granger causal relationships for the first lag of changes in the Bank of England's monetary policy and for the second lag in housing sales. This means that house prices are affected 1 month after the change in monetary policy and 2 months after the change in the volume of housing sales. In addition, other variables appear that have statistical value, although with less precision than those mentioned above (0.05 < *p*-value < 0.1). Again, the monetary policy imposed by the central Bank of England appears at the second lag. In this group of variables, the existence of a Granger causal relationship between changes in the FTSE 100 valuation at the second and third lags on house prices is indicated. This finding is relevant, given that, within the financial variables incorporated into this analysis, there are Granger causal relationships resulting not only from internal monetary policy but also from basic financial factors such as the performance of UK companies on the stock market. Moreover, US monetary policy appears in this group with a relevant Granger causal relationship on house prices at the third lag, indicating that the London housing market may well be exposed to the vagaries of the monetary policies of nations with which there is relevant trade. These results will be verified for their impact on house prices through VAR modelling analysis.

**Table 4.** Granger causality test results. Source: Own elaboration.

| Null Hypothesis Tested | Obs. | F-Statistic | *p*-Value | Lags | Null Hypothesis |
|---|---|---|---|---|---|
| *Pol. rate USA does not Granger Cause Housing Prices* | *279* | *2.2108* | *0.0871* | *3* | *Reject \** |
| Pol. rate UK does not Granger Cause Housing Prices | 279 | 2.0049 | 0.1136 | 3 | Accept |
| Pol. rate eurozone does not Granger Cause Housing Prices | 279 | 0.8622 | 0.4612 | 3 | Accept |
| *FTSE 100 does not Granger Cause Housing Prices* | *279* | *2.2130* | *0.0869* | *3* | *Reject \** |
| Pol. rate China does not Granger Cause Housing Prices | 279 | 1.1177 | 0.3423 | 3 | Accept |
| *Housing sales does not Granger Cause Housing Prices* | *279* | *2.4342* | *0.0652* | *3* | *Reject \** |
| Pol. rate USA does not Granger Cause Housing Prices | 280 | 0.6787 | 0.5081 | 2 | Accept |
| *Pol. rate UK does not Granger Cause Housing Prices* | *280* | *2.8958* | *0.0569* | *2* | *Reject \** |
| Pol. rate eurozone does not Granger Cause Housing Prices | 280 | 1.3881 | 0.2513 | 2 | Accept |
| *FTSE 100 does not Granger Cause Housing Prices* | *280* | *2.7769* | *0.0640* | *2* | *Reject \** |
| Pol. rate China does not Granger Cause Housing Prices | 280 | 1.5855 | 0.2067 | 2 | Accept |
| *Housing sales does not Granger Cause Housing Prices* | *280* | *3.0455* | *0.0492* | *2* | *Reject \** |
| Pol. rate USA does not Granger Cause Housing Prices | 281 | 1.6918 | 0.1944 | 1 | Accept |
| *Pol. rate UK does not Granger Cause Housing Prices* | *281* | *4.1920* | *0.0416* | *1* | *Reject \** |
| Pol. rate eurozone does not Granger Cause Housing Prices | 281 | 1.7465 | 0.1874 | 1 | Accept |
| FTSE 100 does not Granger Cause Housing Prices | 281 | 0.1687 | 0.6816 | 1 | Accept |
| Pol. rate China does not Granger Cause Housing Prices | 281 | 0.7050 | 0.4018 | 1 | Accept |
| Housing sales does not Granger Cause Housing Prices | 281 | 0.0018 | 0.9660 | 1 | Accept |

\* = Granger causality between variables is significant.

Table 5 presents the summary results of the VAR modelling of causal relationships over time between house prices and the other variables analysed. On the one hand, the variables obtained in the Granger causality tests are confirmed, but the importance of Chinese monetary policy rate on house prices in London is added to the first lag. In order of statistical weight, the most relevant variables are UK monetary policy at the first lag and US monetary policy at the third lag. This would confirm the finding that house prices in London are highly sensitive to the decisions taken by both the Bank of England and the US Federal Reserve. In the second order of explanatory significance are home sales at the third lag, Chinese monetary policy at the first lag and the FTSE 100 index at the third lag.

**Table 5.** VAR system of lag order 3 for equation on housing prices. Source: Own elaboration.

| Variables and Lags | Coefficients | Std. Error | t-Ratio | *p*-Value | Flags |
|---|---|---|---|---|---|
| *Constant* | *0.00114302* | *0.000527681* | *2.166* | *0.0312* | ** |
| *Housing Price, lag 1* | *0.455565* | *0.0635833* | *7.165* | *$8.22 \times 10^{-12}$* | *** |
| *Housing Price, lag 2* | *0.410782* | *0.069179* | *5.938* | *$9.32 \times 10^{-9}$* | *** |
| Housing Price, lag 3 | −0.0862846 | 0.06605 | −1.306 | 0.1926 | |
| *Policy Rate UK, lag 1* | *0.0114597* | *0.00259998* | *4.408* | *$1.54 \times 10^{-5}$* | *** |
| *Policy Rate UK, lag 2* | *−0.00695584* | *0.00353382* | *−1.968* | *0.0501* | * |
| Policy Rate UK, lag 3 | 0.000729593 | 0.00391384 | 0.1864 | 0.8523 | |
| Policy Rate USA, lag 1 | 0.00298159 | 0.00199155 | 1.497 | 0.1356 | |
| Policy Rate USA, lag 2 | −0.000946352 | 0.00247969 | −0.3816 | 0.703 | |
| *Policy Rate USA, lag 3* | *−0.00735435* | *0.00272528* | *−2.699* | *0.0074* | *** |
| Policy Rate EURO, lag 1 | −0.00473945 | 0.00475526 | −0.9967 | 0.3199 | |
| Policy Rate EURO, lag 2 | 0.000384436 | 0.00364176 | 0.1056 | 0.916 | |
| Policy Rate EURO, lag 3 | 0.00200074 | 0.00343272 | 0.5828 | 0.5605 | |
| *Policy Rate China, lag 1* | *−0.0621311* | *0.0278124* | *−2.234* | *0.0263* | ** |
| Policy Rate China, lag 2 | −0.0207223 | 0.0216578 | −0.9568 | 0.3396 | |
| Policy Rate China, lag 3 | 0.0310942 | 0.0246697 | 1.26 | 0.2087 | |
| FTSE 100, lag 1 | 0.00544968 | 0.011962 | 0.4556 | 0.6491 | |
| FTSE 100, lag 2 | 0.0158896 | 0.0130004 | 1.222 | 0.2227 | |
| *FTSE 100, lag 3* | *0.026536* | *0.0120834* | *2.196* | *0.029* | ** |
| Housing Sales, lag 1 | $6.63 \times 10^{-5}$ | 0.00238664 | 0.02779 | 0.9778 | |
| Housing Sales, lag 2 | 0.00189091 | 0.00193964 | 0.9749 | 0.3305 | |
| *Housing Sales, lag 3* | *−0.00334143* | *0.00138023* | *−2.421* | *0.0162* | ** |

*** = high statistical significance; ** = good statistical significance; * = statistical significance; = no statistical significance.

The summary statistics of the VAR model presented in Table 6, indicate that there is robustness in the sample, from which conclusions can be drawn that have a basis in reality, beyond the technical exercise involved in the development of this type of study. In other words, the results can be used to rethink the empirical relationship between London's housing policy and the financial factors reviewed here.

**Table 6.** VAR model statistics summary. Source: Own elaboration.

| Model Summary | Values |
|---|---|
| Mean.dependent.var | 0.005834 |
| S.D.dependent.var | 0.009494 |
| Sum.squared.resid | 0.01128 |
| S.E.of.regression | 0.006625 |
| R-squared | 0.54987 |
| Adjusted.R-squared | 0.513089 |
| F(21,257) | 17.12837 |
| *p*-value(F) | $1.25 \times 10^{-37}$ |
| rho | −0.016871 |
| Durbin-Watson | 2.028945 |

The resulting unit root analysis of the model (Table 7) indicates that no root is outside the unit circle, so the VAR model can satisfy the stability condition required for this type of modelling.

**Table 7.** Roots of characteristics polynomial. Source: Own elaboration.

| Root | Modulus |
|---|---|
| 0.791837 | 0.7918367 |
| −0.458833 + 0.599529i | 0.7549585 |
| −0.458833 − 0.599529i | 0.7549585 |
| 0.731298 − 0.163883i | 0.7494359 |
| 0.731298 + 0.163883i | 0.7494359 |
| 0.671551 | 0.6715506 |
| 0.629454 − 0.166120i | 0.6510054 |
| 0.629454 + 0.166120i | 0.6510054 |
| −0.629745 | 0.629745 |
| −0.302615 + 0.468267i | 0.5575387 |
| −0.302615 − 0.468267i | 0.5575387 |
| 0.231456 − 0.472974i | 0.5265701 |
| 0.231456 + 0.472974i | 0.5265701 |
| −0.325271 + 0.328244i | 0.46211 |
| −0.325271 − 0.328244i | 0.46211 |
| 0.115123 + 0.441627i | 0.4563851 |
| 0.115123 − 0.441627i | 0.4563851 |
| −0.121353 + 0.437143i | 0.4536746 |
| −0.121353 − 0.437143i | 0.4536746 |
| −0.314282 | 0.3142817 |
| 0.288777 | 0.2887771 |

Finally, Figure 2 gives relevance to these findings by comparing the impact curves generated by a shock of one standard deviation in the magnitude of these variables on house prices in London. The figure shows positive reactions, which push house prices up, and negative reactions, which push house prices down. Within the magnitudes of comparative

impact, the effect of an FTSE 100 shock has the largest impact on house prices, peaking in the fourth month after the shock. This finding suggests that house prices would be sensitive to the performance of the London Stock Exchange, so there would be an influential relationship between an elementary financial factor such as the FTSE 100 and house prices. A shock to the UK monetary policy rate also generates an impact, which is much more immediate as peaks in the second month, but, as can be seen in Figure 2, the aggregate impact over 10 months is smaller than that of a shock to the FTSE 100. On the other hand, a dual outcome, i.e., pushing up and then pushing down house prices, is observed in shocks induced by US monetary policy and London house sales. In the case of house sales, the effect is alternating, which could be interpreted as indicating that the supply–demand elasticity of house prices tends to seek equilibria. The impact of US monetary policy is interesting, given that it generates an initial positive impact that then tends to level off over time. In other words, a shock in US monetary policy first pushes up house prices and then pushes them down. Shocks in Chinese monetary policy and the eurozone produce reductions in house prices, opening the door to a discussion on the elasticity of house prices in London based on its relationship with its trading partners in terms of imports, given that the eurozone accounts for 56% of imports into the UK, followed in second place by China, which accounts for 12% of imports. This effect based on the relationship with trading partners could be linked to the cost of building materials, but this would require further research.

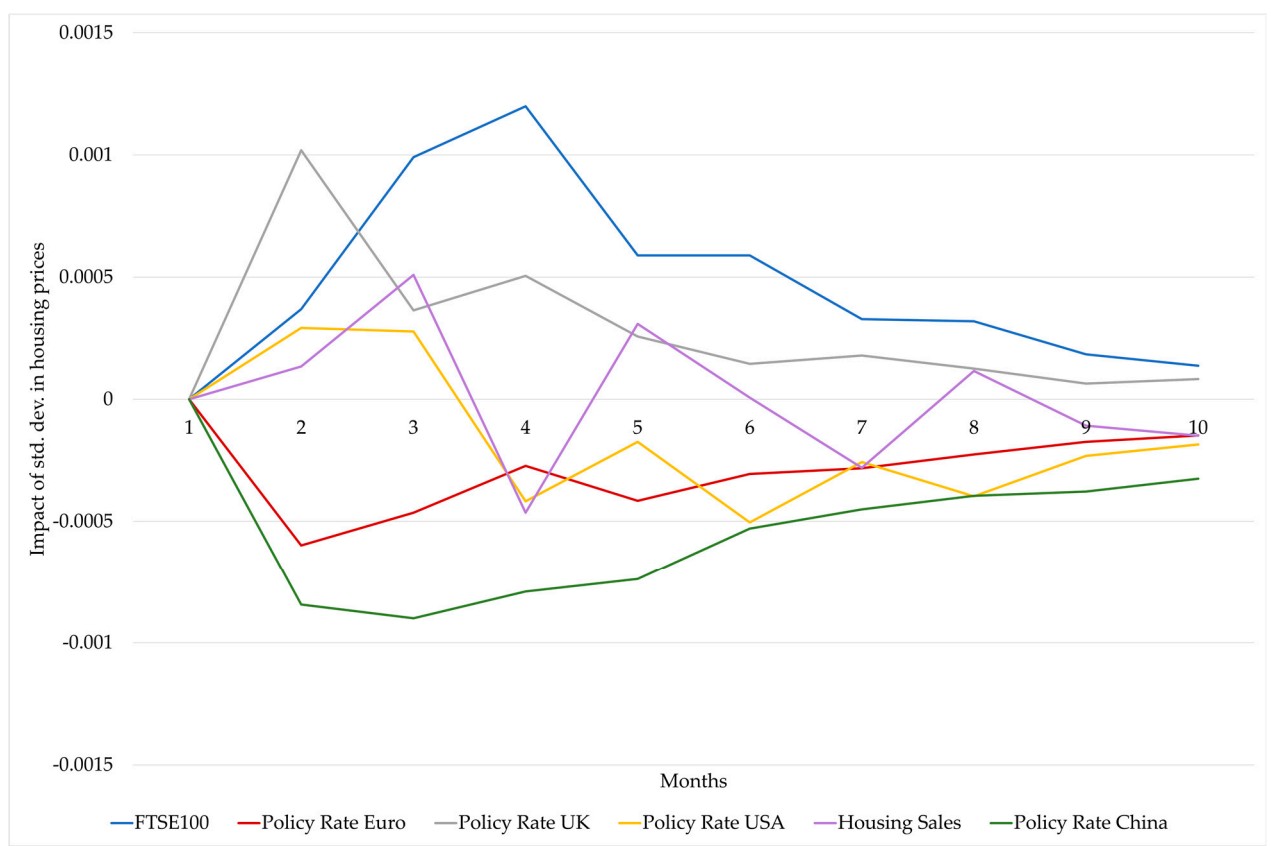

**Figure 2.** Impact on housing prices of a standard deviation shock in each variable based on the results of VAR model. Source: Own elaboration.

## 4. Discussion and Conclusions

The results of this research allow us to advance the discussion on financialisation processes by providing evidence of the probable effect of financial factors on house prices in London. The sensitivity of London house prices to international financial factors such as US or Chinese monetary policy rates indicates a predictive Granger causality between international financial factors and house prices. This may be because London's openness to

markets, being one of the so-called global cities, means that local impacts are also sensitive to variations in these other financial spaces, whose international interconnectedness also affects the household economy. The impact of the FTSE 100 index of the London Stock Exchange on house prices is also significant. This causal relationship in Granger's sense allows us to suggest that the price of housing is indeed influenced upwards by shares traded on the stock exchange, and this confirms the concern of part of the theory of the financialisation of housing, by empirically demonstrating the statistical relationship between these variables.

Concerning the literature that was reviewed in this study, the results allow some specific statements to be made. In part, van der Zwan's argument about the impact of finance on local productive processes is feasible. Furthermore, the results contribute to Kohl's research on the relativity of the effectiveness of market liberalisation in providing better housing solutions. From the findings of this paper, it can be argued that financial openness offers dual outcomes, where some alterations contribute to lowering the price of housing (a desirable objective in terms of affordability) and other alterations in global financial processes tend to increase the price of housing (a desirable objective from the perspective of rentier investors). It is also argued that the empirical evidence presented here suggests that the global expansion of the financial sector is temporally related to housing affordability, especially in societies where access to housing is strongly influenced by the market for the sale and purchase of property, as is the case in London. Moreover, as far as the application of time series models with house prices is concerned, there are also linkages with the reviewed literature. As indicated by Zhang et al., Ibrahim and Law, Kuethe and Pede, and Sá et al., there are interdependent relationships between macroeconomic factors and house prices, which in this case also applies to the London case. Specifically, the impact of base financial elements such as interest rates and the value of shares on the local stock exchange has been measured. However, this study has not explored the effect of general economic and human development on house prices as Yang and Pan did in China, or of immigration as D'Albis et al. did in France. A question that arises from these results is whether the process of housing financialisation recorded in the critical literature has a correlation with the quality of life of people in cities like London (or others of similar global importance) and whether these effects improve or worsen the quality of life. To make these enquiries, new datasets would be needed to complement the findings that have been obtained by testing the impact of these factors on house prices.

London has for years faced a problem associated with securing access to affordable housing. People have adopted a variety of strategies in order to have a place to sleep, ranging from self-purchase for higher-income households to overcrowding. If the process of financialization, measured as it has been in this article, has the effect that has been evidenced in the price of housing, this process of financialisation is also a reproducer of socio-spatial inequality and is therefore a problem for the common good of the city. As the literature review indicates, the process of financialisation intertwines local realities with global markets. This research does not have the necessary data to suggest a solution whereby financial factors could be removed from the processes of housing production and allocation, but it is possible to identify that this is a task for the State, which must assess how to produce effective solutions. This is an open discussion, on which many of the suggestions coming from critical perspectives on the financialisation of housing converge. For now, the situation is a cause for concern, because the solutions will not emerge only from empirical evidence, such as that presented here, but from the capacities of agency that scale up sufficiently to be able to displace the financial power of housing systems, which the financialisation theorists claim is an urgent urban planning matter. The results from London may be replicable in many other cities around the world. There are other financialised territories where the impact on the local economy, reflected in housing prices, could use the same methodology. In addition to testing this model of analysis in other global cities, the study technique could be applied to cities in the global south. This opens up the scope for comparative studies, as long as the quality of the data allows this.

Further research associated with the findings presented in this article can be envisaged. On the one hand, it would be possible to broaden the methodological approach from a purely quantitative one to a mixed one, where statistical data could be blended with interviews and fieldwork to improve the interpretations of these results. On the other hand, it would be relevant to conduct research with a similar methodology in other cities of high global relevance, simultaneously intending to compare whether these findings are really generalisable. One of the limitations of this study is that, even though the statistical significance of financial factors on housing prices has been showed, these factors are not the only ones that have an impact. There are other fundamentals to consider. What has been demonstrated, however, is the validity of the initial hypothesis and that there is a statistically significant relationship to be reviewed. Further scope for future research could be based on results that incorporate other fundamentals of house price formation to see which factors are the most relevant in influencing price. In an article published in the case of Santiago de Chile [37,38], it can be seen that financial factors are statistically significant and more relevant than other house price fundamentals for that city. This is indicative that the research agenda on the econometrics of housing financialisation is broad and compelling.

Finally, the study of the financialisation of housing questions the ethos of urban and architectural practice. If the process of designing, producing and delivering housing is being ordered by the financial world, pursuing profit-driven objectives over providing good living spaces, then the disciplines of urban and architectural design face a crisis of meaning, where they must decide whether to follow the rules of the financial market and thus accept their impact on the work they produce, or whether to sensitise themselves to this situation and move towards a regime of producing good living spaces over rent. There may be paths other than the ones suggested here, but the contradiction exists and demands action from urban practitioners.

**Funding:** This research was funded by Universidad De Las Américas-Chile, through Centro Producción del Espacio and the APC was funded by Vicerrectoría De Investigación, Universidad De Las Américas-Chile.

**Data Availability Statement:** The data presented in this study are available on request from the corresponding author. You can ask the data in the following site: https://www.researchgate.net/publication/370150571_London_data_for_testing_financialisation_of_housing?utm_source=twitter&rgutm_meta1=eHNsLXlJZGxSY1ZxTlBSOThNZmZYOTA0ZmxuMk1hUkVzNGNjN3p5bXE5OUgxS1M0eVVxemVyU1c2am0vN01vUFFCeFk4cjhRWHhURUI3V2xJdjjNtZXVkY1kvST0%3D.

**Conflicts of Interest:** The author declares no conflict of interest.

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
