# Peer review of "Financialisation of Housing in London: Empirical Evidence on Housing Prices"

_urbansci, doi:10.3390/urbansci7020045_

Round 1

Reviewer 1 Report

The economic rationale or theoretical basis of the econometric model use be explained a bit more, preferably by citing similar studies in the past. Also, at the end of the paper it is better to include areas for further exploration of the present research (directions for future research). 

Slight modifications (corrections) in the language (eg. empirical cone-xions, be changes as empirical connections).

Author Response

The economic rationale or theoretical basis of the econometric model use be explained a bit more, preferably by citing similar studies in the past.

RESPONSE: Thank you for this comment. New paragraphs have been added citing other studies that apply similar analysis techniques to house prices and which also allow for a broader discussion of the results.

Also, at the end of the paper it is better to include areas for further exploration of the present research (directions for future research).

RESPONSE: Thank you for this comment. While the last part of the manuscript already presented directions for future research, new ideas have been added to complement those possible expansions of this research.

Slight modifications (corrections) in the language (eg. empirical cone-xions, be changes as empirical connections).

RESPONSE: A throughout revision was run in a proofreading way which may have resolved the issues.

Reviewer 2 Report

This paper examines the financialisation of housing in London, UK. The study shows that housing prices have some relationship with the financial instruments. This is an interesting study.

In my opinion, I am afraid that I am not entirely convinced that the paper, in its present form, represents a clear contribution to knowledge and that there are remaining deficiencies in the paper.  I think the authors are right in general to assert that more research in the area of housing financialization is warranted, and the paper and underlying research have some merits.  But I don't think in the current form that the paper/narrative or the research are sufficiently rigorous for a journal like Urban Science which has high expectations as regards rigour and originality of contribution. The following suggestions hope to assist the authors in revising the paper.

Major Comments:

The motivation of this paper is not entirely convincing and clear. In my opinion, the current paper purely applies a number of variables to examine their relationship with housing price. So, I was not entirely convinced. In any event, this approach builds up the idea that the main contribution is about econometric technique rather than understanding the policy issues relating to the financialization of housing. The motivation of the study should be enhanced. Why we need to have this dedicated study?

The contribution of the study is somewhat weak. The author claim the following as the contribution of this study.

“The study is original, given that this type of modelling has not previously been carried out for a major world city such as London, and contributes to other findings from similar explorations that have applied other methodologies.”

I am not convinced that no such modelling has been placed on major cities. This study basically used different eco variables to explain housing prices. This type of modelling has been widely used previously. For instance, Bangura and Lee (2020) in Housing Studies.

This links to the next point of a more comprehensive literature review. I am afraid that many previous related literature has not been considered in the literature review. The author should try to google scholar with some key words such as “market fundamentals”, “house prices”, “housing price bubble”, and he/she will find many studies in these areas for major cities such as London, Sydney, Hong Kong etc.

These studies employed various variables, including economic variables and social-economic variables, as well as air quality as explanatory variables for housing prices. Importantly, these variables are significant in explaining housing prices. Why these variables are not considered or controlled (e.g., housing supply, air quality, household income etc)? The bottom line is that the author should discuss or qualify why these variables are not considered.

The data section needs some clarification. Table 1 shows the statistics for interest rate in Eurozone. But the analysis does not present the results for the Eurozone interest rate. In fact, the Chinese interest rate was employed. Why? Which one is correct?

The use of the interest rates from different countries has not been explained. Why the use of interest rate of US? The logic of using it very clear, but this should be explained. Unfortunately, no discussion is provided. This is very disappointing.

The methodology section should be clarified. Are these variables I(0) or I(1) or I(2)? This is critical to examine whether these variables are I(0) prior to any modelling. The results in Table 2 seems to show that these variables are I(0). This is surprising as economic variables are rare to be I(0).

In addition, the results in Table 4 can be improved. Instead of running Granger-causality with different lags. I believe the authors should determine the optimal lags first and then examine the Granger-causality. Importantly, the results seem to be varied by using different lags. This also highlights that the results are not robust to different lags, highlighting the importance of identifying the optimal lags.

The discussion of the results can be enhanced. The current   version merely reports the findings instead of discussing the results. For instance, the following discussion from lines 223-226:

“In order of statistical weight, the most relevant variables are UK monetary policy at the first lag and

US monetary policy at the third lag. This would confirm the finding that house prices in London are highly sensitive to the decisions taken by both the Bank of England and the US Federal Reserve.”

So why house prices in London are highly sensitive to rates in the UK and US? How about Chinese and Eurozone interest rates? Some explanation is required to provide a fuller understanding of the housing price dynamics.

Minor comments:

1)      Proof reading is required

2)      Some statistics are not consistent with the discussion. Please double check. For instance, Table 1.

Author Response

The motivation of this paper is not entirely convincing and clear. In my opinion, the current paper purely applies a number of variables to examine their relationship with housing price. So, I was not entirely convinced. In any event, this approach builds up the idea that the main contribution is about econometric technique rather than understanding the policy issues relating to the financialization of housing. The motivation of the study should be enhanced. Why we need to have this dedicated study? The contribution of the study is somewhat weak. The author claim the following as the contribution of this study. “The study is original, given that this type of modelling has not previously been carried out for a major world city such as London, and contributes to other findings from similar explorations that have applied other methodologies.” I am not convinced that no such modelling has been placed on major cities. This study basically used different eco variables to explain housing prices. This type of modelling has been widely used previously. For instance, Bangura and Lee (2020) in Housing Studies.

RESPONSE: Thank you for your comment. We are sorry that the study's motivation was not convincing for the reviewer, and some changes were introduced to try an enhancement in this sense. We have reviewed the reference provided and it confirms the fact that there are no econometric studies linking the theory of financialization with its empirical effects in cities like London regarding housing prices. As a fundamental part of current urban studies, the processes of financialization lack a correlate in the statistical evidence related to housing prices, which this study provides. In this regard, the citation referred to in the text is somewhat out of context, since what the manuscript states in the abstract is:

As a practical matter, these results provide empirical background to pursue this theory more specifically on the vectors that are effectively causal to financialisation processes that impact on everyday life through housing prices. The study is original, given that this type of modelling has not previously been carried out for a major world city such as London, and contributes to other findings from similar explorations that have applied other methodologies.

However, in seeking to pick up on the commentary as a shortcoming of what the article presents, different sections of the manuscript have incorporated new literature that better illustrates the gap in the literature it seeks to fill.

I am afraid that many previous related literature has not been considered in the literature review. The author should try to google scholar with some key words such as “market fundamentals”, “house prices”, “housing price bubble”, and he/she will find many studies in these areas for major cities such as London, Sydney, Hong Kong etc.

RESPONSE: Thank you for your contribution. The authors used WOS and Scopus to search peer-reviewed literature on the matter but there are not many references that articulate VAR modeling with the theory of financialization, none in London. It is important to stress that this articulation takes the critical reflections on financialisation effects on everyday life to test, so framing the exploration in that way is novel in literature. We have expanded the reference base that uses VAR models for housing prices in order to strengthen this section, although none of those references applied a specific framing to critical reflections on financialisation and its eventual effects on housing prices. The new references were mostly included in the introduction section.  

These studies employed various variables, including economic variables and social-economic variables, as well as air quality as explanatory variables for housing prices. Importantly, these variables are significant in explaining housing prices. Why these variables are not considered or controlled (e.g., housing supply, air quality, household income etc)? The bottom line is that the author should discuss or qualify why these variables are not considered.

RESPONSE: Thank you for your comment. I think it is important to reinforce that the study does not aim to test the impact of socioeconomic variables on housing prices, but solely financial variables, as indicated directly in line 118 of the first version of the manuscript, in the methods and data section.

The aim of this paper is to review whether there are causal relationships between financial variables and house prices in this direction of causality:

Financial Variables -> Housing Prices.

The data section needs some clarification. Table 1 shows the statistics for interest rate in Eurozone. But the analysis does not present the results for the Eurozone interest rate. In fact, the Chinese interest rate was employed. Why? Which one is correct? The use of the interest rates from different countries has not been explained. Why the use of interest rate of US? The logic of using it very clear, but this should be explained. Unfortunately, no discussion is provided. This is very disappointing.

RESPONSE: Thank you for your comment. However, the specific observation is not understood since the causal relationships of the Eurozone interest rate are presented in Tables 4 and 5, which are not statistically significant and therefore are discarded as influential variables in the study. Regarding the reason for incorporating interest rates from other regions into the study, it is explained in the existing content of the text, given that London is a city particularly exposed to the international market and trade, so the influence of other markets is relevant, as also cited in the literature in the introduction of the text in row 88, and also in row 128:

Considering that the United Kingdom and London in particular is a city with a great openness to international trade, the monetary policy rate of the Euro-Zone, China and the United States of America was incorporated into the analysis.

The methodology section should be clarified. Are these variables I(0) or I(1) or I(2)? This is critical to examine whether these variables are I(0) prior to any modelling. The results in Table 2 seems to show that these variables are I(0). This is surprising as economic variables are rare to be I(0).

RESPONSE: Thanks for this comment. Table 2 already indicates the statistical descriptors of each variable in use, which are non-stationary based on the ADF tests performed and already reported (also you can check figure 1 to see the curve). Also, Table 7 present the unit root analysis.

In addition, the results in Table 4 can be improved. Instead of running Granger-causality with different lags. I believe the authors should determine the optimal lags first and then examine the Granger-causality. Importantly, the results seem to be varied by using different lags. This also highlights that the results are not robust to different lags, highlighting the importance of identifying the optimal lags.

RESPONSE: Thank you for this comment, however, I am unable to understand the aim of the observation made. Table 3 indicates that, according to the Aikaike criterion, Granger causality should be examined over three lags. This is precisely what Table 4 presents. It is expected that Granger tests are not always robust for each lag because that is precisely what is being verified: how many lags it takes for variables to impact each other, if there is any impact at all. Since there is no impact in some lags, the finding is precisely that the relationship between variables is weak at that specific lag.

The discussion of the results can be enhanced. The current   version merely reports the findings instead of discussing the results. For instance, the following discussion from lines 223-226:

 “In order of statistical weight, the most relevant variables are UK monetary policy at the first lag and

US monetary policy at the third lag. This would confirm the finding that house prices in London are highly sensitive to the decisions taken by both the Bank of England and the US Federal Reserve.”

So why house prices in London are highly sensitive to rates in the UK and US? How about Chinese and Eurozone interest rates? Some explanation is required to provide a fuller understanding of the housing price dynamics.

RESPONSE: Thanks for this comment. An enhanced version of the discussion was now included to address these questions.

Minor comments:

1)      Proof reading is required

RESPONSE: A proofreading was performed in the new version of the manuscript. Thanks for the advice.

2)      Some statistics are not consistent with the discussion. Please double check. For instance, Table 1.

RESPONSE: It would be good what is observed here as the comment is too broad. Perhaps this observation was already responded above. Specifically, what is understood as “consistent with the discussion” in this comment is difficult to understand.  

Reviewer 3 Report

Overall, the article starts off well, but then lacks some critical engagement with the findings. I think the major issue is asserting causation with a method that, while it claims to identify causation, does not truly do this. I think you need to be careful how you approach this and not make such bold claims as that international monetary policy is a direct cause of housing prices in London. Of course, it is a factor, but not a cause alone. You need to do more in your discussion putting the analysis you present in context. You should engage more with housing location theory, and other factors that go into determining housing prices... of which there are many. Why should we believe that your study is more valid than any other? Overall, the paper needs some major revisions to address these major issue. Some more detailed and specific comments are below. Generally, your paper is less than 4000 words, and this is pretty short for a journal article and it shows in the lack of engagement and analysis given.

1. In the introduction, you briefly discuss the London context and some of the housing issues there. I might relate this to some other major cities in order to show that what you are studying in London is relevant to a larger number of cities. It may also be good here to discuss here (or in the discussion) how the UK process of financialization may differ from other European countries or the US.

2. In section 2, paragraph 1 on methods (and other instances in the methods), you state that the aim of this paper is to review whether there are causal relationships between financial variables and house prices. I would specify here that you are attempting to use the Granger causality test (which you will then specify more further down). I would also say that you need to generally be cautious about asserting causation, since as with other statistical analyses, causation cannot truly be determined. I would discuss more the benefits and drawbacks of the Granger causality test and explain more what it is actually testing for. You start to do this towards the end of the methods section, but it can be detailed more, particularly focusing on how it is a "causality test" and what this really means. This also means explaining more what a lag is in this method.

3. You use a VAR model, but never really explain what this is or why you use it. Please expand on this.

4. In your discussion (lines 293-295), you need to cite these comments. Generally, your discussion/conclusion doesn't cite any literature at all, and it probably should. It would also be great to discuss more the implications of "the power to manage solutions moves away from states to global financial power networks." While this may be a part, I think you need to discuss more and in this also talk more about the limitations of this work. Again, this is the issue of causation. The financialization measures you examine are not the sole "causes" of house prices in London, despite what your methods may enable you to say... They are a complex interaction of many many factors... of which the factors you examine may be a part. But you need to lessen your overall assertion on the causation of these factors on housing prices. You then need to expand this discussion of how the need to consider these factors may impact housing policy at the city and national scale. Ultimately, saying the state has no power to manage this is essentially saying the state has no power in the national housing market, which is not true.

Author Response

  1. In the introduction, you briefly discuss the London context and some of the housing issues there. I might relate this to some other major cities in order to show that what you are studying in London is relevant to a larger number of cities. It may also be good here to discuss here (or in the discussion) how the UK process of financialization may differ from other European countries or the US.

RESPONSE: Thank you for this comment. New paragraphs have been added citing other studies that apply similar analysis techniques to house prices and which also allow for a broader discussion of the results.

  1. In section 2, paragraph 1 on methods (and other instances in the methods), you state that the aim of this paper is to review whether there are causal relationships between financial variables and house prices. I would specify here that you are attempting to use the Granger causality test (which you will then specify more further down). I would also say that you need to generally be cautious about asserting causation, since as with other statistical analyses, causation cannot truly be determined. I would discuss more the benefits and drawbacks of the Granger causality test and explain more what it is actually testing for. You start to do this towards the end of the methods section, but it can be detailed more, particularly focusing on how it is a "causality test" and what this really means. This also means explaining more what a lag is in this method.

RESPONSE: Thank you for your comment. Indeed, in its previous version the article lacked the care to refer to causal relationships in Granger's sense, exclusively. This new version includes a warning about its use in the article and explains that when it talks about causality it refers to Granger's model, specifically.

  1. You use a VAR model, but never really explain what this is or why you use it. Please expand on this.

RESPONSE: Thank you for your comment. New content was added that situates the use of the VAR method in the house price literature. This should help to better situate the article in the choice of this model which is mentioned in the methods and data section again.

  1. In your discussion (lines 293-295), you need to cite these comments. Generally, your discussion/conclusion doesn't cite any literature at all, and it probably should. It would also be great to discuss more the implications of "the power to manage solutions moves away from states to global financial power networks." While this may be a part, I think you need to discuss more and in this also talk more about the limitations of this work. Again, this is the issue of causation. The financialization measures you examine are not the sole "causes" of house prices in London, despite what your methods may enable you to say... They are a complex interaction of many many factors... of which the factors you examine may be a part. But you need to lessen your overall assertion on the causation of these factors on housing prices. You then need to expand this discussion of how the need to consider these factors may impact housing policy at the city and national scale. Ultimately, saying the state has no power to manage this is essentially saying the state has no power in the national housing market, which is not true.

RESPONSE: Thank you for your comment. Indeed, there was too much grandiloquence in the last section of the article. It has been revised and changes have been implemented to avoid presenting such overconfident arguments when indeed the results illustrate a specific aspect of house price formation and not its actual totality.

Round 2

Reviewer 3 Report

Comments from previous review have all been addressed adequately.

Author Response

Thank you for accepting this manuscript with minor revisions. The text has been thoroughly reviewed to correct its grammatical errors and to ensure consistency in the use of British spelling. 
